# Learning variable-length skills through Novelty-based Decision Point Identification

## Abstract

Intelligent agents are able to make decisions based on different levels of granularity and duration. Recent advances in skill learning with data-driven behavior priors enabled the agent to solve complex, long-horizon tasks by effectively guiding the agent in choosing appropriate skills. However, the practice of using fixed-length skills can easily result in skipping valuable decision points, which ultimately limits the potential for further exploration and faster policy learning. For example, making a temporally-extended decision at a crossroad can offer more direct access to parts of the state space that would otherwise be challenging to reach. In this work, we propose to learn variable-length skills by identifying decision points through a state-action novelty module that leverages offline agent experience datasets, which turns out to be an efficient proxy for the critical decision point detection. We show that capturing critical decision points can further accelerate policy learning by enabling a more efficient exploration of the state space and facilitating transfer of knowledge across various tasks. Our approach, NBDI (Novelty-based Decision Point Identification)[1], substantially outperforms previous baselines in complex, long-horizon tasks (e.g. robotic manipulation and maze navigation), which highlights the importance of decision point identification in skill learning.

## 1 Introduction

The ability to make decisions based on different levels of granularity and duration is one of the key attributes of intelligence. In reinforcement learning (RL), temporal abstraction refers to the concept of an agent reasoning over a long horizon, planning, and taking high-level actions. Each high-level action corresponds to a sequence of primitive actions, or low-level actions. For example, in order to accomplish a task with a robot arm, it would be easier to utilize high-level actions such as grasping and lifting, instead of controlling every single joint of a robot arm. Temporal abstraction simplifies complex tasks by reducing the number of decisions the agent has to make, thereby alleviating the challenges that RL faces in long-horizon, sparse reward tasks.

Due to the advantages of temporal abstraction, there has been active research on developing hierarchical RL algorithms, which structure the agent's policy into a hierarchy of two policies: a high-level policy and a low level policy. The option framework (Sutton, 1998) was proposed to achieve temporal abstraction by learning options, which are high-level actions that contain inner low level policy, initiation set and termination conditions. Termination conditions are used to figure out when to switch from one option to another, enabling the agent to flexibly respond to changes in environment or task requirements. While the option framework can achieve temporal abstraction without any loss of performance when the options are optimally learned, it is usually computationally challenging to optimize for the ideal set of options within complex domains.

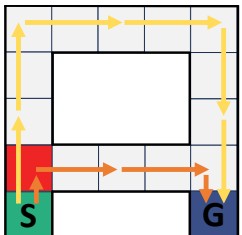

Figure 1: An example where discovering fixed-length skills is highly inefficient.

In this case, the skill discovery framework, which aims to discover meaningful skills (fixed-length executions of low-level policy) from the dataset through unsupervised learning techniques, has been

---

[1]Our code is available at: `https://github.com/asdfnbdi/nbdi`

used as an alternative. Recently, notable progress has been made in skill-based deep RL models, showing promising results in complex environments and robot manipulations (Pertsch et al., 2021a; Hakhamaneshi et al., 2021). However, the use of fixed-length skills and the absence of appropriate termination conditions often restrict them from making decisions at critical decision points (e.g., crossroads). This can result in significant loss in performance, as illustrated in Figure 1. While there have been some studies incorporating the option framework into deep RL as is, the algorithmic complexity and unstable performance in large environments limit its widespread adoption (Kulkarni et al., 2016; Hutsebaut-Buysse et al., 2022).

In this paper, we present NBDI (Novelty-based Decision Point Identification), a task-agnostic, simple state-action novelty-based decision point identification method that allows the agent to learn variable-length skills through critical decision point detection. Identifying critical decision points promote knowledge transfer between different tasks and stimulate exploration by closely connecting different areas in the state space (McGovern & Barto, 2001; Menache et al., 2002; Şimşek & Barto, 2004). For example, detecting doorways between rooms is useful regardless of the specific task at hand. We demonstrate the straightforward applicability of our method to the skill-based deep RL framework, and illustrate how it can lead to improvements in decision-making.

The paper is organized as follows: we first introduce the discovery of state-action novelty-based critical decision points in reinforcement learning (Section 4). Next, we demonstrate how we learn variable-length skills through state-action novelty (Section 5). Then we illustrate the inefficiency of employing fixed-length skills and demonstrate that executing variable-length skills, based on state-action novelty, can accelerate policy learning in both robot manipulation and navigation tasks (Section 6). Finally, we provide insights into how our model successfully uses state-action novelty to improve policy learning by implementing several ablation studies (Appendix A).

## 2 RELATED WORK

**Option Framework**  One major approach of discovering good options is to focus on identifying good terminal states, or sub-goal states. For example, landmark states (Kaelbling, 1993), reinforcement learning signals (Digney, 1998), graph partitioning (Menache et al., 2002; Şimşek et al., 2005; Machado et al., 2017a;b), and state clustering (Srinivas et al., 2016) have been used to identify meaningful sub-goal states. Digney (1998); Simsek et al. (2005) and Kulkarni et al. (2016) focused on detecting bottleneck states, which are states that appear frequently within successful trajectories, but are less common in unsuccessful trajectories (e.g., a state with access door). Şimşek & Barto (2004) tried to identify access states, which are similar to bottleneck states, but determined based on the relative novelty score of predecessor states and successor states. Access states are found based on the intuition that sub-goals will exhibit a relative novelty score distribution with scores that are frequently higher than those of non sub-goals. These studies motivated us to search for states with meaningful properties to terminate skills. However, these methods frequently face challenges in scaling to large or continuous state spaces.

**Skill-based deep RL**  As extending the classic option framework to high-dimensional state spaces through the adoption of function approximation is not straightforward, a number of practitioners have proposed acquiring skills, which are fixed-length executions of low-level policies, to achieve temporal abstraction. For example, skill discovery (Gregor et al., 2016; Achiam et al., 2018; Mavor-Parker et al., 2022) and skill extraction (Yang et al., 2021; Singh et al., 2020; Pertsch et al., 2021b; Hakhamaneshi et al., 2021) frameworks have proven to be successful in acquiring meaningful sets of skills. Especially, Pertsch et al. (2021a) showed promising results in complex, long-horizon tasks with sparse rewards by extracting skills with data-driven behavior priors. The learned prior enables the agent to explore the environment in a more structured manner, which leads to better performance in downstream tasks. However, we believe that its performance is greatly constrained by the use of fixed-length skills, which restricts them from making decisions at critical decision points.

**Novelty-based RL**  Novelty has been utilized in reinforcement learning for various purposes. Depending on its design, novelty can be used for curiosity-driven exploration (Burda et al., 2018; Pathak et al., 2019; Sekar et al., 2020), or data coverage maximization (Bellemare et al., 2016; Hazan et al., 2019; Seo et al., 2021). It has been also used to identify sub-goals in discrete environments (Goel, 2003; Şimşek & Barto, 2004). However, to the best of our knowledge, there has

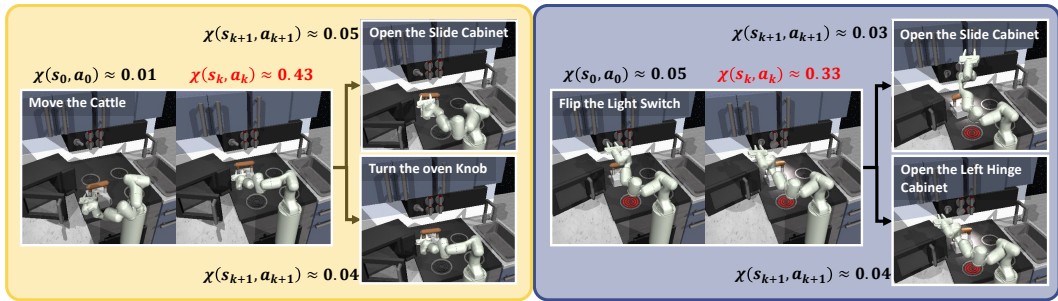

Figure 2: Visualization of an example of critical decision points in the kitchen environment. High state-action novelty can be found in states where a subtask has been completed, and multiple subsequent subtasks are accessible.

been no research that has utilized state-action novelty for identifying decision points in the context of deep RL or for improving exploration in downstream tasks.

## 3 BACKGROUND

**Markov Decision Process (MDP)** MDP is a mathematical framework to model decision making problems with discrete-time control processes. It is defined by a tuple $\{\mathcal{S}, \mathcal{A}, P, R\}$, where $\mathcal{S}$ denotes a state space, $\mathcal{A}$ denotes a set of actions the agent can execute, $P(s'|s, a)$ denotes a transition probability and $R(s, a)$ is a reward function. In a MDP, the probability of transitioning to a future state depends solely on the current state, which is known as the Markov property. Given a MDP, we aim to find an optimal policy $\pi^*$ that maximizes the expected discounted sum of reward. The state value function $V^\pi(s)$ and the action value function $Q^\pi(s, a)$ denote the conditional expectation of discounted sum of reward following policy $\pi$.

**Option Framework** The option framework (Sutton, 1998) is one of the first studies to achieve temporal abstraction in RL. The option framework is composed of two major elements: a meta-control policy $\mu$ and a set of options $\mathcal{O}$. An option is defined as $\langle \mathcal{I}, \pi, \beta \rangle$, where $\mathcal{I} \subseteq \mathcal{S}$ defines an initiation set, $\pi : \mathcal{S} \times \mathcal{A} \to [0, 1]$ defines a policy, and $\beta : \mathcal{S} \to [0, 1]$ defines a termination condition. The policy $\pi$ chooses the next action, until the option is terminated by the stochastic termination condition $\beta$. Once the option terminates, the agent has an opportunity to switch to another available option at the termination state. Options usually refer to low-level polices that are promised to be good only for a subset of the state space. Thus, the presence of an appropriate initiation set $\mathcal{I}$ and termination condition $\beta$ is crucial for the agent's overall performance.

Any MDP with a fixed set of options can be classified as a Semi-Markov Decision Process (SMDP) (Sutton, 1998). SMDP (Bradtke & Duff, 1994) is an extended version of MDP for the situations where actions have different execution lengths. It serves as the foundational mathematical framework for many hierarchical RL algorithms, including the option framework.

## 4 SIMPLE AND EFFICIENT IDENTIFICATION OF DECISION POINTS

The option framework aims to achieve temporal abstraction by learning good options, and good options can be learned through the identification of meaningful sub-goal states (Digney, 1998; Menache et al., 2002; Şimşek & Barto, 2004), i.e., the critical decision points. In this work, we propose to use state-action novelty to identify critical decision points for skill termination, which leads to the execution of variable-length skills. Compared to other approaches for decision point identification, our proposed method is much simpler to implement, and it can be used jointly with any skill-based hierarchical RL algorithms. Furthermore, any state-action novelty estimation mechanism that measures the joint novelty of state-action pairs can be used for our approach.

## 4.1 STATE-ACTION NOVELTY-BASED DECISION POINT IDENTIFICATION

In short, our proposed method classifies a state-action pair with high joint state-action novelty as a decision point. A more insightful perspective on this choice can be obtained by breaking down a novelty estimator as in (1). By interpreting joint novelty $\chi(s,a)$ as the reciprocal of joint visitation count $N(s,a)$, we can decompose a state-action joint novelty $\chi$ into the product of a state novelty and a conditional action novelty. The proposed method combines the strength of both novelty estimates.

$$\chi(s,a) = \frac{1}{N(s,a)} = \frac{1}{N(s)} \cdot \frac{1}{N(a|s)} = \underbrace{\chi(s)}_{\text{state novelty}} \cdot \underbrace{\chi(a|s)}_{\text{conditional action novelty}} \tag{1}$$

The state novelty $\chi(s)$ will seek for a *novel state*, which refers to a state that is either challenging to reach or rare in the dataset of agent experiences. As the skills are derived from the same pool of experiences that we use to estimate the novelty, a high state novelty implies a potential lack of diverse skills to explore neighboring states effectively. Increasing the frequency of decision-making in such unfamiliar states will lead to improved exploration and broader coverage of the state space.

A conditional action novelty $\chi(a|s)$ will seek for a *novel action*. With the state conditioning, action novelty will be high in a state where a *multitude of actions* have been frequently executed. For example, unlike straight roads, crossroads provide the agent with options to move in multiple directions. In such states, the agent may need to perform different actions to accomplish the current goal, rather than solely depending on the current skill. This necessity arises because the current skill may have been originally designed for different goals, making it potentially less than ideal for the current goal. Guiding the agent to make more decisions in such states can increase the likelihood of solving the task at hand, ultimately accelerating the policy learning.

**Examples** Critical decision points are not just limited to navigation tasks; they can also be found in robot manipulation tasks. In the kitchen environment, as shown in Figure 2, high state-action novelty $\chi(s,a)$ tends to occur in states where a subtask has been completed. After completing the subtask of flipping the light switch, the agent has the option to open either the left hinge cabinet or the right slide cabinet. In sequential manipulation tasks, such critical points are valuable because completing one subtask grants access to multiple other subtasks.

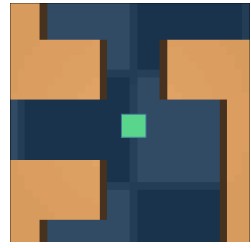

Figure 3: An example of critical decision points in the maze environment that has diverse plausible actions (crossroad).

Figure 3 shows an example of critical decision points in the maze environment. When the agent encounters a maze with an unseen goal, it would have no way of knowing which direction would lead to the goal. Therefore, encouraging the agent to make more decisions at crossroads would effectively connect different areas within the maze, ultimately promoting exploration. As a multitude of actions tends to be executed at crossroads, the state-action novelty $\chi(s,a)$ tends to be high in these states due to the high conditional action novelty $\chi(a|s)$.

## 4.2 TERMINATION IMPROVEMENT FROM STATE-ACTION NOVELTY-BASED TERMINATIONS

We provide an alternative interpretation on the potential benefits of identifying decision points based on state-action novelty. While maximizing skill length is advantageous in terms of temporal abstraction, extended skills can result in suboptimal behavior, especially when the skills are derived from task-agnostic trajectories. Such suboptimality of extended skills (or options) can be theoretically quantified using the termination improvement theorem (Sutton, 1998).

**Theorem. [Termination Improvement, Sutton (1998), informal]** *For any meta-control policy $\mu$ on set of options $\mathcal{O}$, define a new set of options $\mathcal{O}'$, which is a set of options that we can additionally choose to terminate whenever the value of a state $V^\mu(s)$ is larger than the value of a state given that we keep the current option $o$, $Q^\mu(s,o)$. With $\mu'$, which has the same option selection probability as $\mu$ but over a new set of options $\mathcal{O}'$, we have $V^{\mu'}(s) \geq V^\mu(s)$.*

The termination improvement theorem basically implies that we should terminate an option when there are much better alternatives available from the current state. When the options/skills are dis-

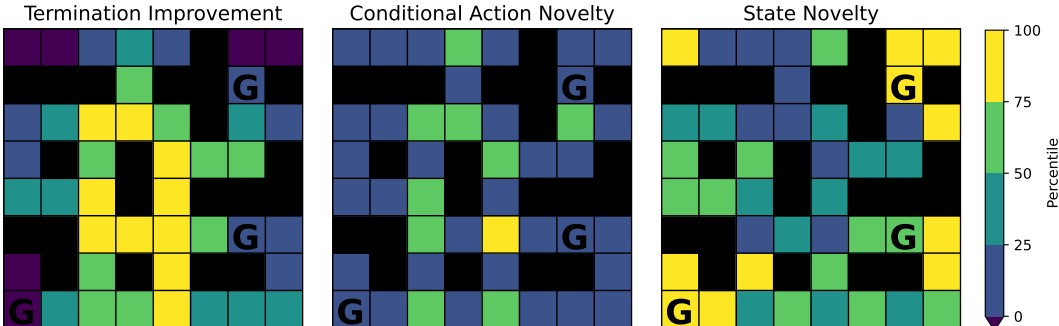

Figure 4: The relative frequency of termination improvement occurrences **(left)**, conditional action novelty **(middle)**, and state novelty **(right)** in a small grid with three different goals. Higher percentile colors indicate a relatively greater number of termination improvement occurrences, higher conditional action novelty, and higher state novelty. Further details on the visualization procedure are provided in Appendix H.1

covered from diverse trajectories (e.g., trajectories gathered from a diverse set of goals), termination improvement is typically observed in states where a multitude of actions have been executed, such as crossroads.

To identify the states where termination improvement occurs, we plotted the relative frequency of termination improvement occurrences in a small grid maze with three different goal settings (Figure 4 (left)). It shows that termination improvement frequently occurs in states where diverse plausible actions exist. In states with a single available option, $V^\mu(s)$ would be equal to $Q^\mu(s,o)$. On the other hand, as more actions/options are plausible, $Q^\mu(s,o)$ would exhibit a broader range of values, thereby increasing the likelihood of satisfying $Q^\mu(s,o) < V^\mu(s)$.

However, terminating skills based on the termination improvement theorem can be challenging when the downstream task is unknown, as it requires $Q^\mu(s,o)$ and $V^\mu(s)$ to be computed in advance with the skills extracted from the downstream task trajectories. Thus, by leveraging the data collected across a diverse set of tasks, we propose to employ conditional action novelty as a tool for pinpointing the states where a multitude of plausible actions can be taken (Figure 4 (middle)). We have also found state novelty to be useful in terminating skills, as it encourages the agent to sufficiently explore unfamiliar parts of the state space (Figure 4 (right)).

## 5 Learning Variable-length Skills through Novelty-based Decision Point Identification

Our goal is to accelerate the learning of a new complex, long-horizon task by deriving variable-length skills from a state-action novelty module. While fixed-length skills have been mostly considered for temporal abstractions in recent studies (Pertsch et al., 2021a; Hakhamaneshi et al., 2021), utilizing fixed-length skills can easily skip valuable decision points, ultimately reducing the opportunities for further exploration and faster policy learning.

In this work, we propose to incorporate state-action novelty into the skill prior and skill embedding learning procedure to effectively capture critical decision points and execute skills of variable-lengths. Our approach, as shown in Figure 5, consists of three major steps: (i) Training the state-action novelty model, (ii) Learning the skill prior, skill embedding space and termination distribution with the pre-trained novelty model, (iii) Performing reinforcement learning with skills of variable-length to solve an unseen task.

**Problem Formulation**    For training the skill prior, skill embedding space, and state-action novelty module, we assume access to unstructured agent experiences of states and actions in the form of $N$ trajectories $\mathcal{D} = \left\{ \tau^i = \{(s_t, a_t)\}_{t=0}^{T-1} \right\}_{i=0}^{N-1}$, which are collected across a diverse set of tasks except for the one we are specifically interested in. Since we do not make any assumptions about rewards

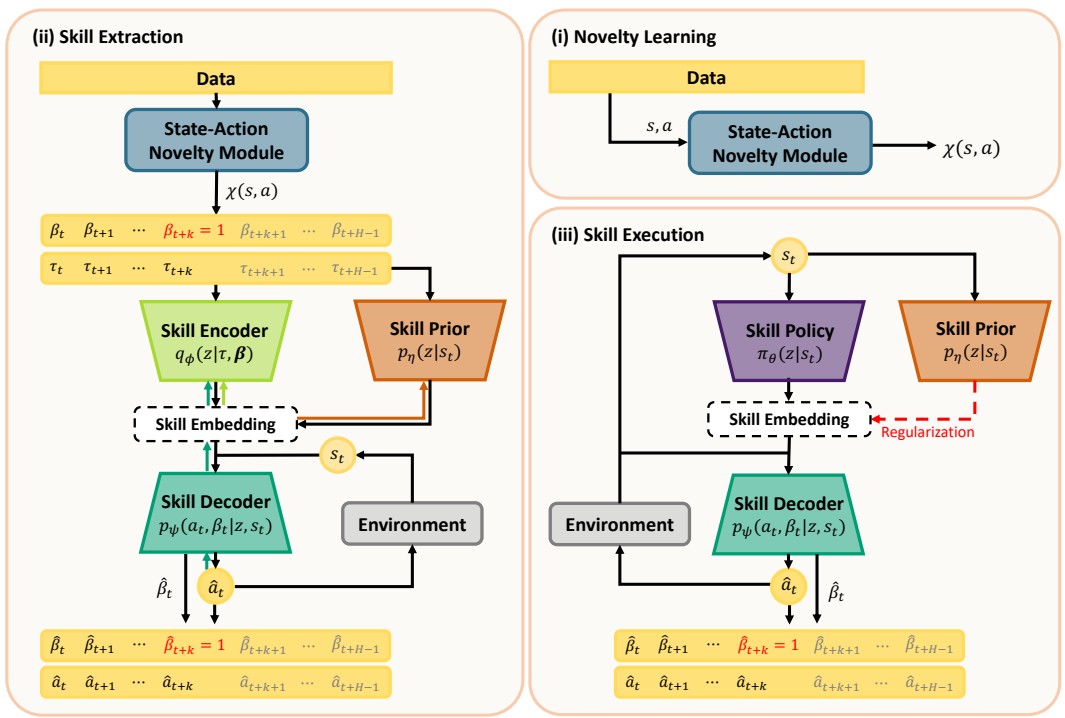

Figure 5: Our approach, Novelty-based Decision Point Identification (NBDI), has three main procedures: (i) **novelty learning**: training the state-action novelty model. (ii) **skill extraction:** learning the skill prior, skill embedding space and termination distribution with the pre-trained novelty model. (iii) **skill execution**: performing reinforcement learning with skills of variable-length to solve an unseen task.

or task labels, our model can leverage real-world datasets that can be collected at a lower cost (e.g., autonomous driving and drones).

## 5.1 UNSUPERVISED LEARNING OF VARIABLE-LENGTH SKILLS

In the process of unsupervised learning, our goal is to pre-train the termination distribution, the skill latent space and the skill prior. We define a skill $z \in \mathcal{Z}$ as an embedding of state-action pairs $\tau = \{(s_i, a_i)\}_{i=t}^{t+H-1}$ and termination conditions $\boldsymbol{\beta} = \{\beta_i\}_{i=t}^{t+H-1}$. The termination conditions $\beta$ are Bernoulli random variables that decide when to stop the current skill. Through the classification of state-action pairs demonstrating significant novelty $\chi(s, a)$, $\beta$ are trained to predict the critical decision points. The point at which novelty is considered significant varies depending on the environment. In downstream tasks, the skill being executed will be terminated either when $\beta = 1$ is sampled or when the maximum skill length $H$ is reached.

To learn the skill embedding space $\mathcal{Z}$, we train a latent variable model consisting of a Long short-term memory (LSTM) (Hochreiter & Schmidhuber, 1997) encoder $q_\phi(z|\tau, \boldsymbol{\beta})$ and a decoder $p_\psi(a_t, \beta_t|z, s_t)$. To learn model parameters $\phi$ and $\psi$, the latent variable model receives a randomly sampled experience $\tau$ from the training dataset $\mathcal{D}$ along with a termination condition vector $\boldsymbol{\beta}$ from the state-action novelty module, and tries to reconstruct the corresponding action sequence and its length (i.e., point of termination) by maximizing the evidence lower bound (ELBO):

$$\log p(a_t, \beta_t|s_t) \geq \mathbb{E}_{z \sim q_\phi(z|\tau, \boldsymbol{\beta}), \tau \sim \mathcal{D}} [\underbrace{\log p_\psi(a_t, \beta_t|z, s_t)}_{\mathcal{L}_{\text{rec}}(\phi, \psi)} + \alpha \underbrace{(\log p(z) - \log q_\phi(z|\tau, \boldsymbol{\beta}))}_{\mathcal{L}_{\text{reg}}(\phi)}] \quad (2)$$

where $\alpha$ is used as the weight of the regularization term (Higgins et al., 2016). The Kullback-Leibler (KL) divergence between the unit Gaussian prior $p(z) = \mathcal{N}(0, I)$ and the posterior $\log q_\phi(z|\tau, \boldsymbol{\beta})$ makes smoother representation of skills.

---

**Algorithm 1** Reinforcement learning with variable-length skills

---

1: **Inputs:** trained skill decoder $p_\psi(a, \beta|z, s)$, discount factor $\gamma$, target divergence $\delta$, learning rates $\lambda_\pi, \lambda_Q, \lambda_\omega$, target update rate $\epsilon$
2: Initialize replay buffer $\mathcal{D}$, high-level policy $\pi_\theta(z|s)$, critic $Q_\xi(s, z)$, target network $\bar{\xi} = \xi$
3: **for** each iteration **do**
4:     **for** each environment step **do**
5:         $z_t \sim \pi_\theta(z_t|s_t)$         ▷ sample skill from policy
6:         **for** $k = 0, 1, \ldots$ **do**
7:             $a_{t+k}, \beta_{t+k} \sim p_\psi(a_{t+k}, \beta_{t+k}|z_t, s_{t+k})$
8:             $s_{t+k+1} \sim p(s_{t+k+1}|s_{t+k}, a_{t+k})$     ▷ execute skill in environment
9:             **if** $\beta_{t+k} = 1$ **then** Break
10:         $\tilde{r}_t \leftarrow \sum_{i=t}^{t+k} \gamma^{i-t} R(s_i, a_i)$         ▷ compute $k$-step reward
11:         $\mathcal{D} \leftarrow \mathcal{D} \cup \{s_t, z_t, \tilde{r}_t, s_{t+k+1}, k\}$     ▷ store transition in replay buffer
12:     **for** each gradient step **do**
13:         $z_{t+k+1} \sim \pi_\theta(z_{t+k+1}|s_{t+k+1})$
14:         $\bar{Q} = \tilde{r}_t + \gamma^k \left[Q_{\bar{\xi}}(s_{t+k+1}, z_{t+k+1}) - \omega D_{KL}(\pi_\theta(z_{t+k+1}|s_{t+k+1})\|p_\eta(z_{t+k+1}|s_{t+k+1}))\right]$
15:         $\theta \leftarrow \theta - \lambda_\pi \nabla_\theta \left[Q_\xi(s_t, z_t) - \omega D_{KL}(\pi_\theta(z_t|s_t)\|p_\eta(z_t|s_t))\right]$
16:         $\phi \leftarrow \xi - \lambda_Q \nabla_\xi \left[\frac{1}{2}(Q_\xi(s_t, z_t) - \bar{Q})^2\right]$
17:         $\omega \leftarrow \omega - \lambda_\omega \nabla_\omega \left[\omega \cdot (D_{KL}(\pi_\theta(z_t|s_t)\|p_\eta(z_t|s_t)) - \delta)\right]$
18:         $\bar{\xi} \leftarrow \epsilon\xi + (1 - \epsilon)\bar{\xi}$
19: **return** trained policy $\pi_\theta(z_t|s_t)$

---

To offer effective guidance in selecting skills for the current state, the skill prior $p_\eta(z|s_t)$, parameterized by $\eta$, is trained by minimizing its KL divergence from the predicted posterior $q_\phi(z|\tau, \boldsymbol{\beta})$. In the context of the option framework, it can also be viewed as the process of obtaining an appropriate initiation set $\mathcal{I}$ for options/skills. This will lead to the minimization of the prior loss:

$$\mathcal{L}_{\text{prior}}(\eta) = \mathbb{E}_{\tau \sim \mathcal{D}} \left[D_{KL}(q_\phi(z|\tau, \boldsymbol{\beta})\|p_\eta(z|s_t))\right] \tag{3}$$

The basic architecture for skill extraction and skill prior follows prior works (Pertsch et al., 2021a; Hakhamaneshi et al., 2021), which have proven to be successful. In summary, termination distribution, skill embedding space, and skill prior are jointly optimized with the following loss:

$$\mathcal{L}_{\text{total}} = \mathcal{L}_{\text{rec}}(\phi, \psi) + \alpha\mathcal{L}_{\text{reg}}(\phi) + \mathcal{L}_{\text{prior}}(\eta) \tag{4}$$

## 5.2 REINFORCEMENT LEARNING WITH VARIABLE-LENGTH SKILLS

In downstream learning, our objective is to learn a skill policy $\pi_\theta(z|s_t)$ that maximizes the expected sum of discounted rewards, parameterized by $\theta$. The pre-trained decoder $p_\psi(a_t, \beta_t|z, s_t)$ decodes a skill embedding $z$ into a series of actions, which persists until the skill is terminated by the predicted termination condition $\beta_t$. The downstream learning can be formulated as a SMDP which is an extended version of MDP that supports actions of different execution lengths.

Adapted from Soft Actor-Critic (SAC) (Haarnoja et al., 2018), we aim to maximize discounted sum of rewards while minimizing its KL divergence from the pre-trained skill prior on SMDP. The regularization weighted by $\omega$ effectively reduces the size of the skill latent space the agent needs to explore.

$$J(\theta) = \mathbb{E}_\pi \left[\sum_{t \in \mathcal{T}} \tilde{r}(s_t, z_t) - \omega D_{\text{KL}}\big(\pi(z_t|s_t), p_\eta(z_t|s_t)\big)\right] \tag{5}$$

where $\mathcal{T}$ is set of time steps where we execute skills, i.e., $\mathcal{T} = \{0, k_0, k_0 + k_1, k_0 + k_1 + k_2, \ldots\}$ where $k_i$ is the variable skill length of $i$-th executed skill.

To handle actions of different execution lengths, the following Q-function objective is used:

$$J_Q(\xi) = \mathbb{E}_{(s_t, z_t, \tilde{r}_t, s_{t+k+1}, k) \sim \mathcal{D}, z_{t+k+1} \sim \pi_\theta(\cdot|s_{t+k+1})} \left[\frac{1}{2}(Q_\xi(s_t, z_t) - \bar{Q})^2\right], \tag{6}$$

where   $\bar{Q} = \tilde{r}_t + \gamma^k[Q_{\bar{\xi}}(s_{t+k+1}, z_{t+k+1}) - \omega D_{KL}(\pi_\theta(z_{t+k+1}|s_{t+k+1})\|p_\eta(z_{t+k+1}|s_{t+k+1}))]$

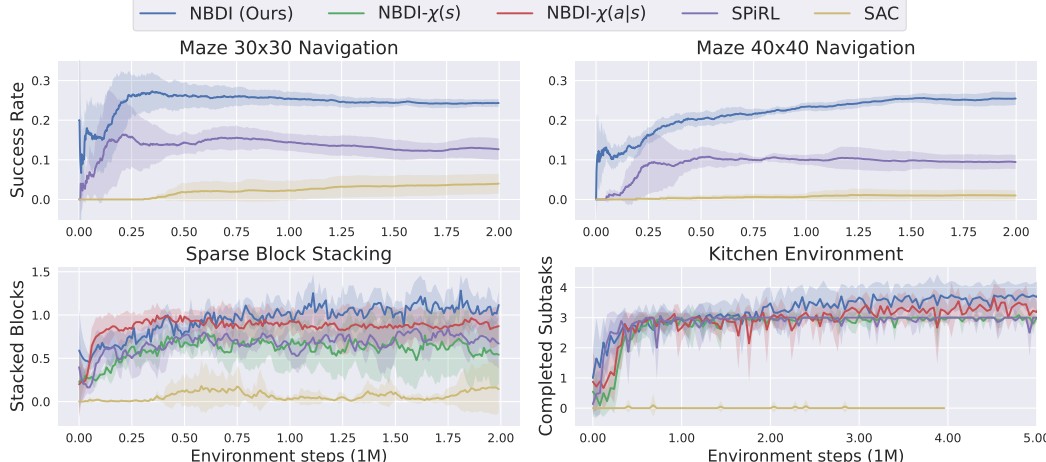

Figure 6: Performances of our method and baselines in solving downstream tasks. The shaded region represents 95% confidence interval across five different seeds.

| Environment | SAC | SPiRL | NBDI-$\chi(a\|s)$ | NBDI-$\chi(s)$ | NBDI (Ours) | Improvement over SPiRL(%) |
|---|---|---|---|---|---|---|
| Maze 30x30 *(Success rate)* | 0.04±0.03 | 0.13±0.03 | 0.07±0.01 | 0.04±0.01 | **0.24**±0.01 | 84.62 |
| Maze 40x40 *(Success rate)* | 0.01±0.01 | 0.09±0.02 | 0.02±0.01 | 0.02±0.02 | **0.25**±0.02 | 177.78 |
| Sparse Block Stacking *(Stacked Blocks)* | 0.14±0.28 | 0.67±0.29 | 0.87±0.19 | 0.54±0.34 | **1.12**±0.16 | 67.16 |
| Kitchen Environment *(Completed Subtasks)* | 0.0±0.0 | 3.0±0.0 | 3.25±0.41 | 3.0±0.0 | **3.67**±0.43 | 22.33 |

Table 1: Performances of our method and baselines in solving downstream tasks. (final performance and 95% confidence interval)

$\omega$ represents the temperature for KL-regularization, $k$ denotes the number of time steps elapsed from the start state $s_t$ to the termination state $s_{t+k+1}$, and $\tilde{r}$ represents the cumulative discounted reward over the $k$ time steps. The detailed RL learning loop is described in Algorithm 1.

## 6  EXPERIMENTS

We designed the experiments to address the following questions: (i) Does learning variable-length skills through critical decision point identification accelerate policy learning in unseen tasks? (ii) How does each component of state-action novelty contribute to the identification of critical decision points? (iii) Have we successfully identified the decision points that match our intuition? In the following experiments, we utilize Intrinsic Curiosity Module (ICM) (Pathak et al., 2017) to calculate state-action novelty for both image-based and non-image-based observations (See Appendix B).

### 6.1  ENVIRONMENTS

A navigation task (mazes sized $30 \times 30$ and $40 \times 40$), and two simulated robot manipulation tasks (kitchen, sparse block stacking) are used to evaluate the performance of NBDI. A large set of task-agnostic agent experiences is collected from each environment to pre-train the termination distribution, skill embedding space, and skill prior. We evaluate the models based on their ability to solve unseen tasks in each specific environment. Further details about the environments and the data collection procedure are provided in Appendix G.

### 6.2  RESULTS

We use the following models for comparison: **Flat RL (SAC)**: Baseline Soft Actor-Critic (Haarnoja et al., 2018) agent that does not leverage prior experience for skill learning. This comparison il-

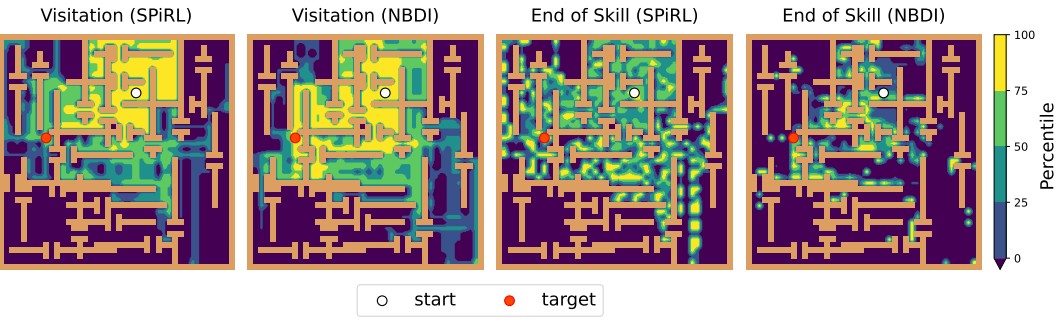

Figure 7: Visualization of decision points made by SPiRL and NBDI in the maze environment. We sampled 100 trajectories for each trained policy to observe the points at which they make decisions. Higher percentile colors suggest a relatively greater number of visitation frequencies and termination frequencies. Note that the termination frequencies are normalized by the overall visitation frequencies for better visualization.

lustrates the effectiveness of temporal abstraction. **Fixed-length Skill Policy (SPiRL)**: The agent that learns a fixed-length skill policy (Pertsch et al., 2021a). This comparison shows the benefit of learning variable-length skills through state-action novelty. **NBDI (Ours)**: The agent that learns a variable-length skill policy through state-action novelty $\chi(s, a)$. All NBDI agents learn the termination distribution $p_\psi(\beta|z, s)$ from each novelty module to predict skill termination at the current step. For robot manipulation tasks, we additionally tested NBDI agents with different types of novelty. **State novelty decision point identification (NBDI-$\chi(s)$)**: The agent that learns a variable-length skill policy through state novelty. To exclusively assess the influence of the novelty type, we distilled the state-action novelty module used in NBDI into a separate network, $\chi(s)$, which solely depends on the current state. **Conditional action novelty decision point identification (NBDI-$\chi(a|s)$)**: The agent that learns a variable-length skill policy through conditional action novelty $\frac{\chi(s,a)}{\chi(s)}$, where $\chi(s)$ is the distilled state novelty module used for NBDI-$\chi(s)$.

As shown in Figure 6, our key findings are as follows: (i) In both the robot manipulation tasks and the navigation task, executing variable-length skills through state-action novelty (NBDI-$\chi(s, a)$) speeds up policy learning and facilitates convergence toward a more effective policy. It can be seen that conditional action novelty (NBDI-$\chi(a|s)$) is also helpful in terminating skills. (ii) In alignment with our motivation for state-action novelty, conditional action novelty appears to play a crucial role in identifying critical decision points. While it appears that terminating skills solely based on state novelty doesn't lead to much performance enhancement, combining it with conditional action novelty leads to better exploration and better convergence. Table 1 indicates that NBDI surpasses SPiRL, even within a challenging robotic simulation environment where there are no clearly defined subtasks (Sparse block stacking). Figure 7 compares critical decision points made by SPiRL and our method in the maze environment. This result provides the answer to our third question: (iii) While the SPiRL agent makes decisions in random states, our model tends to make decisions in crossroad states or states that are unfamiliar. For instance, in the lower-right area of the maze, SPiRL shows periodic skill terminations due to its fixed-length of skills, whereas our approach tends to make decisions in states characterized by high conditional action novelty or state novelty.

## 7 CONCLUSION

We present NBDI, an approach for learning variable-length skills by detecting decision points through a state-action novelty module that leverages offline, task-agnostic datasets. We propose an efficient method that jointly optimizes the termination distribution, skill embedding space, and skill prior using a deep latent variable model. Our approach significantly outperforms prior baselines in solving complex, long-horizon tasks, which highlights the importance of decision point identification in skill learning. A promising direction for future work is to use novelty-based decision point identification to learn variable-length skills in offline execution settings (Ajay et al., 2020; Hakhamaneshi et al., 2021) or in meta-reinforcement learning (Nam et al., 2022).

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

# A ABLATION

## A.1 ABLATION IN VARIABLE-LENGTH SKILLS

Figure 9 (left) compares the performance of our model (NBDI-th0.3) in the kitchen environment with different state-action novelty threshold values. We can see that there is no significant improvement in performance compared to SPiRL when the threshold value is not appropriately chosen. For example, as illustrated in Figure 8, termination distributions learned with low threshold values can disturb the policy learning by terminating skills in states that lack significance. It illustrates that threshold value needs to be appropriately chosen to capture meaningful decision points.

## A.2 ABLATION IN NO TERMINATION DISTRIBUTION

Figure 9 (middle) shows the performance drop when we do not learn the termination distribution in advance. NBDI-NoTermDistr directly uses the state-action novelty module in the downstream learning phase to terminate skills. The performance gap indicates that the skill embedding space needs to be learned with terminated skills to effectively guide the agent in choosing variable-length skills. Thus, it is necessary to jointly optimize the termination distribution, skill embedding space, and skill prior using the deep latent variable model.

Figure 8: A bad example of decision points in the maze environment.

## A.3 ABLATION IN CRITERIA TO DETERMINE DECISION POINTS

Figure 9 (right) shows the performance difference when we use cumulative sum of state-action novelties to learn decision points. NBDI-CumulativeSum terminates skills once the cumulative sum of state-action novelty reaches or surpasses a predefined threshold. This comparison implies that accumulating novelties does not lead to the identification of significant termination points.

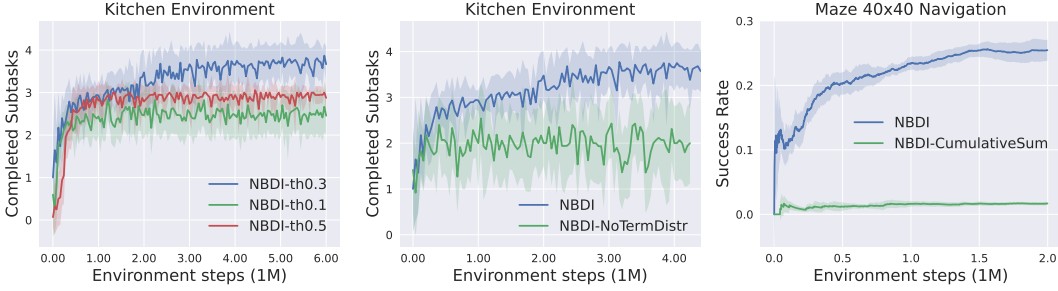

Figure 9: Ablation in variable-length skills **(left)**, no termination distribution **(middle)**, and criteria to determine decision points **(right)**. The shaded region represents 95% confidence interval across five different seeds.

# B USING INTRINSIC CURIOSITY MODULE TO ESTIMATE STATE-ACTION NOVELTY

We use Intrinsic Curiosity Module (ICM) to calculate state-action novelty for both image-based and non-image-based observation environments. While ICM is typically recognized for providing intrinsic motivation signals to drive exploration in online RL, we found it to be an efficient state-action novelty estimator when it is pre-trained with offline trajectory datasets. Since ICM takes in state-action pair to predict next state representation, it would have high prediction error for sparse state-action pairs in the offline dataset.

Figure 10 illustrates the prediction error of state-action pairs from 25 randomly selected trajectories within the offline trajectory dataset used for training ICM. We visualized the states of the state-action pairs with high prediction error ((A), (B), (C)) and low prediction error ((D), (E), (F)) on the right. It can be seen that high prediction error can be typically seen in states where we have multitude of plausible actions ((A), (C)) or rare state configuration (B). Note these characteristics correspond to the conditional action novelty and state novelty as illustrated in Figure 4, and leads to a high state-action novelty as in Equation 1. On the other hand, low prediction error can be seen in states where we do not have any of these properties ((D), (E), (F)). Since a stack of two consecutive images is used as state for training ICM and NBDI, it is easy for ICM to get the next state prediction correct in these states. Figure 11 shows prediction error of state-action pairs in sparse block stacking environment, which is a more complex robotic simulation environment that has no clearly defined subtasks. In this environment, the agent needs to stack blocks on top of each other. We can see that high prediction error occurs in states where the agent has multitude of plausible actions ((A), (B), (C)). When the robotic arm is positioned above a block, it must choose between descending to lift that block or moving towards other blocks. Likewise, when the robotic arm is holding a block, it needs to determine which block to stack it onto. This characteristic also corresponds to the conditional action novelty as depicted in Figure 4, and results in a high state-action novelty as in Equation 1. Similar to the maze environment, low prediction error occurs in states where we do not have multitude of plausible actions, or rare configuration ((D), (E), (F)). Thus, our findings validate that ICM can serve as an efficient state-action novelty estimator when pre-trained with offline trajectory datasets.

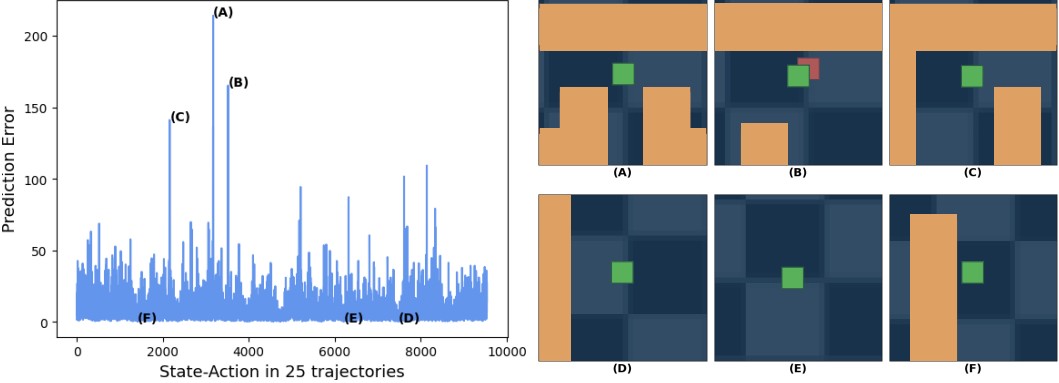

Figure 10: Visualization of prediction error of ICM in maze environment. Note the same offline data that is used to train ICM was used to compute this prediction error. (A), (B) and (C) are the state-action pairs with the highest prediction error, while (D), (E) and (F) are the ones with the lowest.

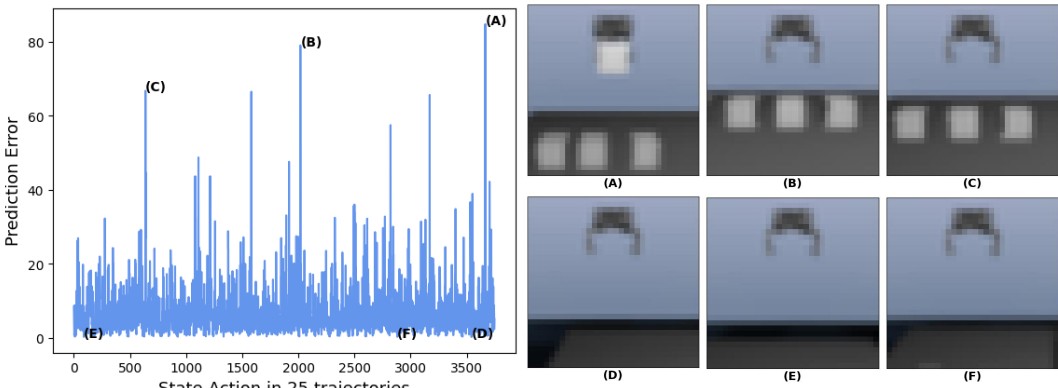

Figure 11: Visualization of prediction error of ICM in sparse block stacking environment. Note the same offline data that is used to train ICM was used to compute this prediction error. (A), (B) and (C) are the state-action pairs with the highest prediction error, while (D), (E) and (F) are the ones with the lowest.

## C    CRITICAL DECISION POINTS WITH SUBOPTIMAL DATA

We investigate how the quality of offline data affects the performance of our approach. We trained a behavior cloning (BC) policy on expert-level trajectories to generate mediocre quality demonstrations. We additionally added weighted Gaussian random noise to actions of BC policy to add stochasticity to the generated dataset. Table 2 shows that even in a less challenging goal setting compared to Figure 7, SPiRL fails to reach the goal, while NBDI achieves a success rate of 28% and 22%. Figure 12 shows that SPiRL can only navigate around the initial state using fixed-length skills extracted from the suboptimal dataset, whereas NBDI can successfully reach the goal by employing variable-length skills. Figure 13 (top, middle) shows that with suboptimal dataset, NBDI is still able to learn termination points characterized by high conditional action novelty or state novelty. However, with dataset generated by random walk (Figure 13 (bottom)), it becomes challenging to learn meaningful decision points. As the policy generating the trajectory becomes more stochastic, it gathers data primarily around the initial state, leading to an overall reduction in the scale of prediction errors. Thus, we can see that the level of stochasticity influences critical decision point detection.

| Dataset quality | NBDI | SPiRL |
|---|---|---|
| Stochastic BC ($\sigma = 0.5$) | **28**% | 0% |
| Stochastic BC ($\sigma = 0.75$) | **22**% | 0% |

Table 2: Success rate of NBDI and SPiRL with offline trajectories generated by mediocre-level policy with weighted Gaussian random noise in maze environment

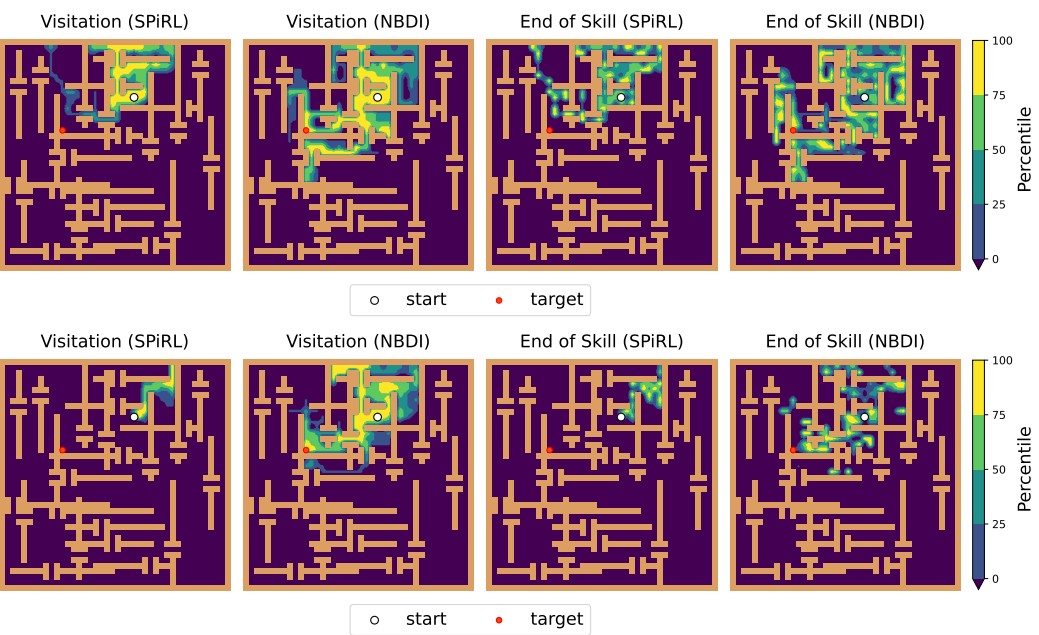

Figure 12: Visualization of visitations and decision points made by SPiRL and NBDI in the maze environment (**top**: trained with stochastic BC ($\sigma = 0.5$) dataset, **bottom**: trained with stochastic BC ($\sigma = 0.75$) dataset). We sampled 100 trajectories for each trained policy to observe the points at which they make decisions. Higher percentile colors suggest a relatively greater number of visitation frequencies and termination frequencies. Note that the termination frequencies are normalized by the overall visitation frequencies for better visualization.

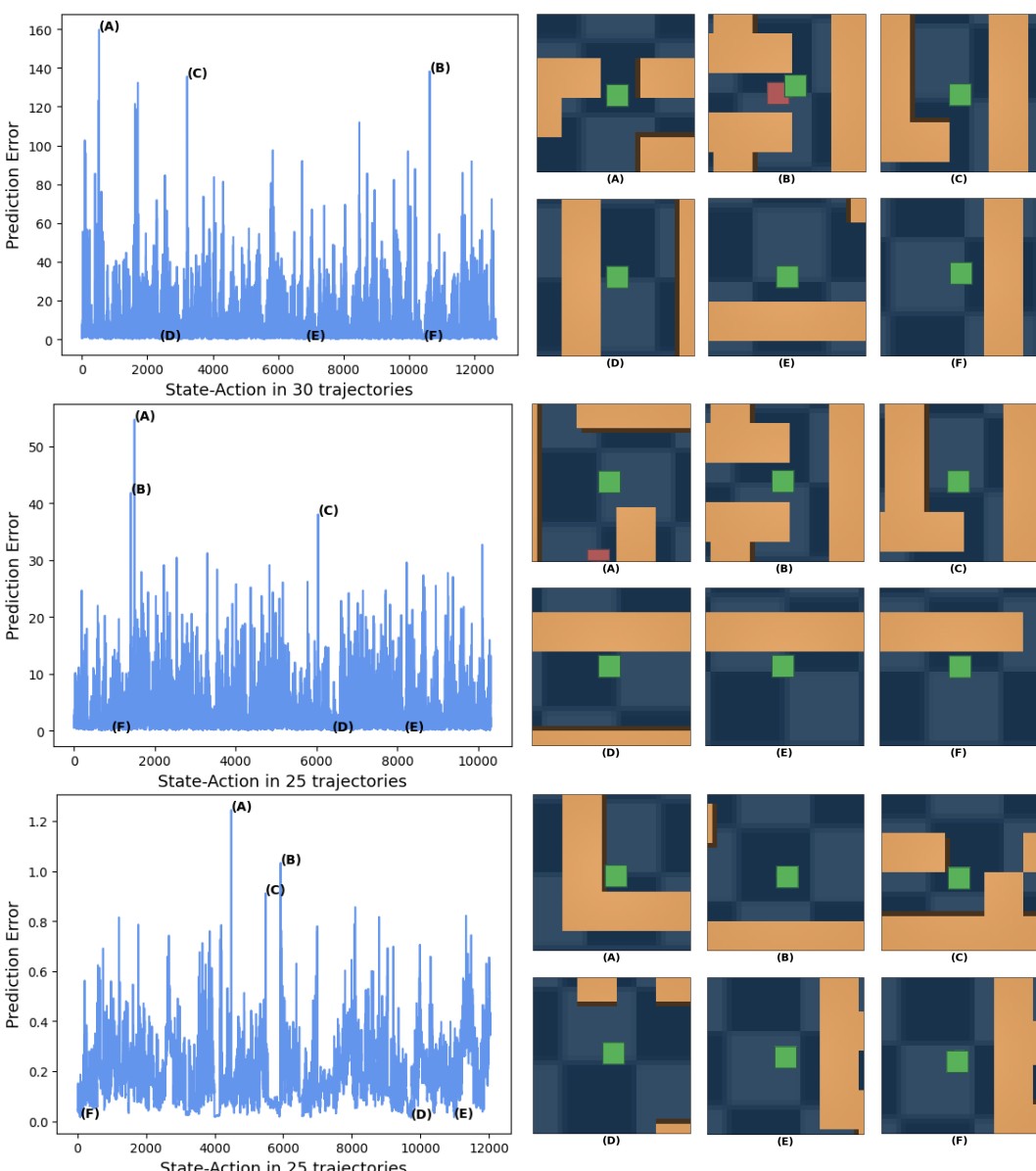

Figure 13: Visualization of prediction error of ICM in maze environment (**top**: trained with stochastic BC ($\sigma = 0.5$) dataset, **middle**: trained with stochastic BC ($\sigma = 0.75$) dataset, **bottom**: trained with random walk dataset). Note the same offline data that is used to train ICM was used to compute this prediction error. (A), (B) and (C) are the state-action pairs with the highest prediction error, while (D), (E) and (F) are the ones with the lowest.

# D    CRITICAL DECISION POINTS IN COMPLEX PHYSICS SIMULATION TASKS

We further investigate whether meaningful decision points can be found in complex physics simulation tasks. We trained ICM using different offline datasets provided by D4RL (Fu et al., 2021) (halfcheetah-medium-expert, halfcheetah-medium-replay, ant-medium-expert, ant-medium-replay) to assess its ability to detect critical decision points. Figure 14 illustrates the presence of critical decision points in complex physics simulation tasks. For instance, the cheetah has the option of spreading its hind legs or lowering them to the ground, and the ant has the choice of flipping to the right or lowering themselves to the ground. However in completely random datasets (halfcheetah-random, ant-random), we were not able to find any meaningful decision points. Similar to Appendix C, it shows that the degree of stochasticity present in the offline dataset can influence critical decision point detection.

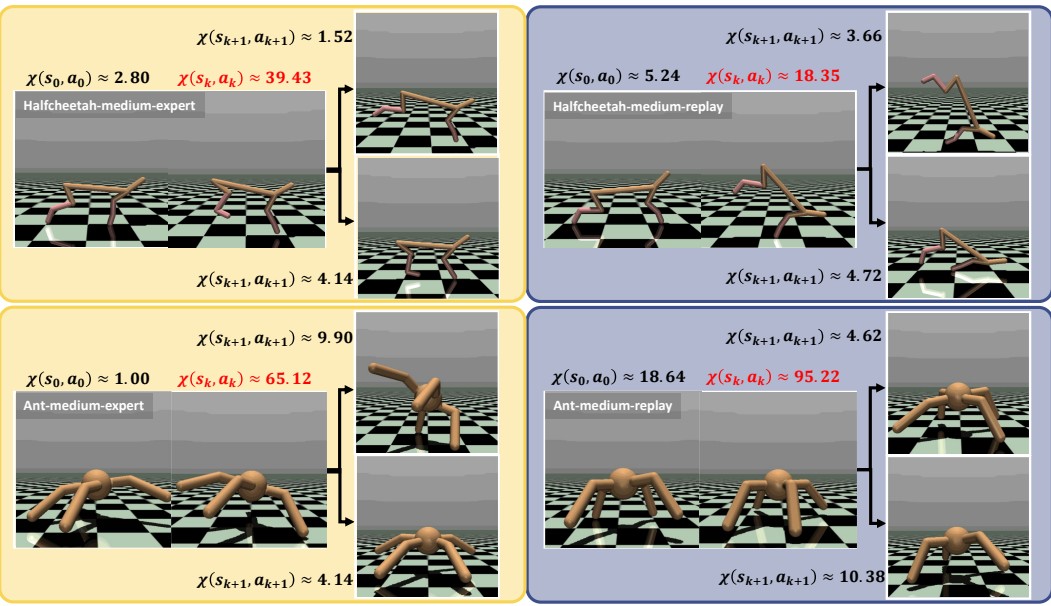

Figure 14: Visualization of critical decision points in MuJoCo (Todorov et al., 2012) environment (**top-left**: halfcheetah-medium-expert, **top-right**: halfcheetah-medium-replay, **bottom-left**: ant-medium-expert, **bottom-right**: ant-medium-replay)

## E  MODEL CAPACITY AND CRITICAL DECISION POINT DETECTION

To assess how the model capacity and dataset utilization influence critical point detection, we varied the width and depth of the neural network across different settings. Figure 15 (left) shows that the estimated state-action novelty is not affected by the number of parameters used to train the model. Figure 15 (right) demonstrates that the scale of the estimated state-action novelty remains consistent even as the dataset size decreases. We can see that our proposed approach for detecting critical decision points is robust to various number of parameters or size of datasets.

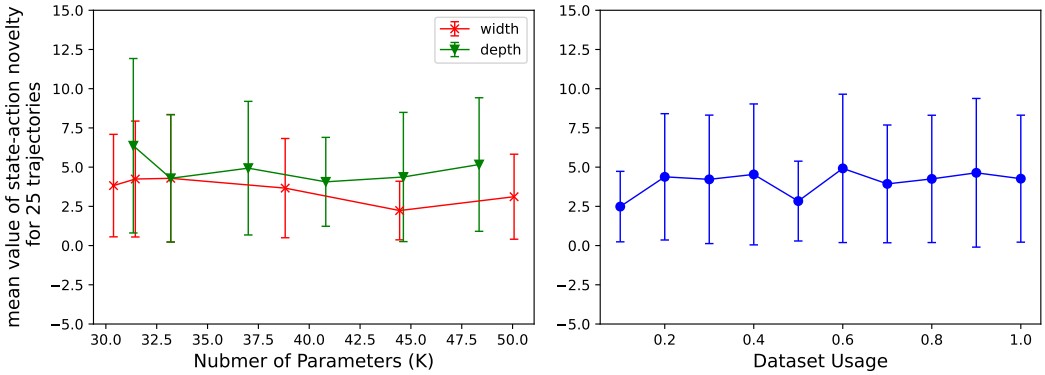

Figure 15: Illustration of the impact of varying the number of parameters and dataset usage on state-action novelty estimation. The error bar represents the standard deviation of state-action novelty across 25 trajectories.

## F  COMPARISON TO DIFFERENT VARIABLE LENGTH ALGORITHM

We compared the performance of NBDI and LOVE (Learning Options Via Compression) (Jiang et al., 2022) in the maze environment. LOVE extracts variable-length skills from the offline experience dataset with an objective that integrates the maximum likelihood objective while imposing a penalty on the length of skill descriptions.

Figure 16 shows that LOVE encounters difficulties in learning useful variable-length skills in complex maze environment. Since LOVE relies on consistent structures within the dataset to effectively learn variable-length skills (Jiang et al., 2022), it faces challenges in learning variable-length skill in complex environments.

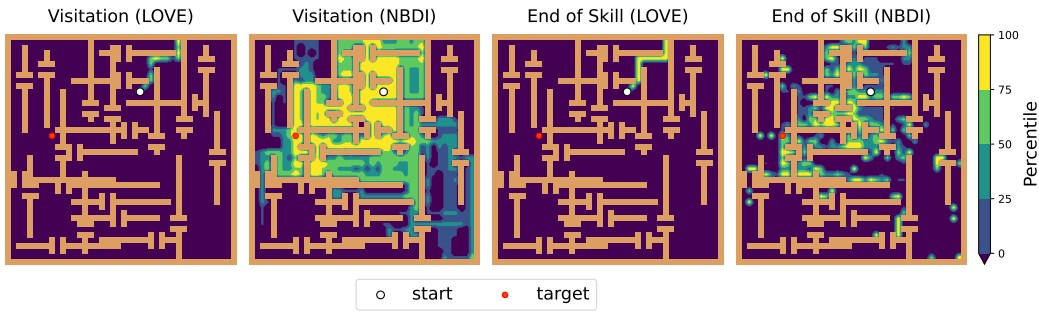

Figure 16: Visualization of visitations and decision points made by LOVE and NBDI in the maze environment. We sampled 50 trajectories for each trained policy to observe the points at which they make decisions. Higher percentile colors suggest a relatively greater number of visitation frequencies and termination frequencies.

## G   DATA AND ENVIRONMENT DETAILS

We evaluate NBDI on three environments: one simulated navigation task (maze navigation) and two simulated robotic manipulation tasks (kitchen and sparse block stacking). We employ the environment configuration and dataset provided by Pertsch et al. (2021a). Note that task and environment setup differ between the training data and the downstream task, demonstrating the model's capacity to handle *unseen* downstream tasks.

**Kitchen Environment**   The kitchen environment is provided by the D4RL benchmark (Fu et al., 2021), featuring seven manipulable objects. The training trajectories consist of sequences of object manipulations. The downstream task of the agent involves performing an unseen sequence of four object manipulations.

The agent is tested by its ability to reassemble the skills learned from the training dataset to solve the downstream task.

- State space: 30-dimensional vector of the agent's joint velocities and the positions of the manipulatable objects

- Action space: 7-dimensional set for controlling robot joint velocities and a 2-dimensional set for gripper opening/closing degree

- Reward: one-time reward upon successfully completing any of the subtasks

**Maze Navigation**   The maze navigation environment is derived from the D4RL benchmark (Fu et al., 2021). During the collection of training data, a maze is generated randomly, and both the starting and goal positions are selected at random as well. The agent successfully reaches its goal in all of the collected trajectories.

In the downstream task, the maze layout is four times bigger than the one employed during training.

- State space: $(x, y)$-velocities and an image of local top-down view centered around the agent

- Action space: $(x, y)$-directions

- Reward: binary reward when the agent's position is close to the goal (computed using Euclidean distance)

**Sparse Block Stacking**   The sparse block stacking environment is created using the Mujoco physics engine. To gather training data, a hand-coded data collection policy interacts with a smaller environment with five blocks to stack as many blocks as possible.

In the downstream task, the agent's objective is to stack as many blocks as possible in a larger version of the environment with eleven blocks.

- State space: $(x, z)$-displacements for the robot

- Action space: 10-dimensional continuous symmetric gripper movements

- Reward: only rewarded for the height of the highest stacked blocks

**Differences to Pertsch et al. (2021a).**   While Pertsch et al. (2021a) employed a block stacking environment with dense rewards (the agent is rewarded based on the height of the stacked tower and for actions like picking up or lifting blocks), we evaluated our model and the baselines in a sparse block stacking environment, leading to different performance outcomes. In this setting, the agent is rewarded solely for the height of the tower it constructs, which increases the task complexity. Furthermore, there have been consistent reports indicating that performance in large maze environments displays a high level of sensitivity to random seeds, primarily due to the high stochasticity of the task. For the five different seeds that we used to compare the algorithms, we found the performance to be generally lower than what was previously reported, mainly due to the sensitivity to seeds.

# H    IMPLEMENTATION DETAILS

## H.1    TERMINATION IMPROVEMENT AND NOVELTY

We present termination improvement and conditional action novelty in a simple $8 \times 8$ grid maze domain. As we mentioned in Appendix G, training data for the state-action novelty module is collected with diverse tasks. Thus, we set up the environment as follows: We randomly select one starting location and three goal locations to generate three trajectories for each goal location. The goal-reaching data collection policy randomly executes a discrete action, moving towards four different directions (left, right, forward, backward) while avoiding moving toward walls. The agent receives a binary reward when reaching the goal state.

We define an option $o = \langle \mathcal{I}, \pi, \beta \rangle$ where a deterministic policy $\pi$ follows the given trajectory, an initiation set $\mathcal{I} \subseteq \mathcal{S}$ defines all states that the policy visits, and a termination condition $\beta$ defines states where the option terminates (every option has a length of three). Each set of options, denoted as $\mathcal{O}_g$ for each goal $g = 1, 2, 3$, contains distinguishable options for each goal location. The frequencies of termination improvement occurrences in each state for each goal setting have been aggregated to generate Figure 4.

Using the trajectories collected from different goals, state novelty and conditional action novelty are simply computed as $\frac{1}{N(s)}$ and $\frac{N(s)}{N(s,a)}$ respectively. $N(s)$ represents the number of times a discrete state $s$ has been visited and $N(s,a)$ represents the number of times a discrete state-action pair has been used.

## H.2    STATE-ACTION NOVELTY MODULE

We use Intrinsic Curiosity Module (ICM) to calculate state-action novelty for both image-based and non-image-based observations. The feature encoder $\phi$, responsible for encoding a state $s_t$ into its corresponding features $\phi(s_t)$, is implemented differently for each environment. In the kitchen environment, it consists of a single fully-connected layer with a hidden dimension of 120. The both maze and sparse block stacking environments have three convolution layers with (4, 4) kernel sizes and (8, 16, 32) channels.

The forward dynamic model $f$ takes $a_t$ and $\phi(s_t)$ as inputs to predict the feature encoding of the state at time step $t + 1$. In the kitchen environment, the structure of the dynamic model is the same as its feature encoder. In the maze and sparse block stacking environment, a single fully-connected layer with hidden dimension 52 and 70 have been used, respectively.

The state-action novelty $\chi(s, a)$ is computed as the squared L2 distance between $\hat{\phi}(\phi(s_t), a_t)$ and $\phi(s_{t+1})$, representing the prediction error in the feature space. We employed the Adam optimizer with $\beta_1 = 0.9, \beta_2 = 0.999$ and a learning rate of $1e-3$ to train the ICM. We found that state-actions within the top 1% prediction error percentile serve well as a critical decision points, and used the corresponding threshold to learn the termination distribution.

## H.3 HYPERPARAMETERS

| Hyperaparameter | Value |
|---|---|
| **State-Action Novelty Module** | |
| Batch size | 150 |
| Optimizer | Adam($\beta_1 = 0.9, \beta_2 = 0.999, lr = 1e-3$) |
| **Kitchen** | |
|   Feature encoder | |
|     hidden dim | 120 |
|   Forward dynamic model | |
|     hidden dim | 120 |
| **Maze** | |
|   Feature encoder | |
|     kernel size | (4, 4) |
|     channels | 8, 16, 32 |
|   Forward dynamic model | |
|     hidden dim | 52 |
| **Sparse Block Stacking** | |
|   Feature encoder | |
|     kernel size | (4, 4) |
|     channels | 8, 16, 32 |
|   Forward dynamic model | |
|     hidden dim | 70 |
| **Skill Prior Learning** | |
| Batch size | 16 |
| Optimizer | RAdam($\beta_1 = 0.9, \beta_2 = 0.999, lr = 1e-3$) |
| Regularization weight $\alpha$ | 1.0 |
| **Skill Encoder** | |
|   dim-$\mathcal{Z}$ in VAE | 32 |
|   hidden dim | 128 |
|   # LSTM Layers | 1 |
| **Skill Prior (Kitchen)** | |
|   hidden dim | 128 |
|   # FC Layers | 6 |
| **Skill Prior (Maze, Block Stacking)** | |
|   kernel size | (4, 4) |
|   channels | 8, 16, 32 |
|   # Convolution Layers | 3 |
| **Skill Decoder** | |
|   hidden dim | 128 |
|   # hidden layers | 6 |
| **Downstream Reinforcement Learning** | |
| Batch size | 256 |
| Optimizer | Adam($\beta_1 = 0.9, \beta_2 = 0.999, lr = 3e-4$) |
| Replay buffer size | 1e6 |
| Discount factor $\gamma$ | $0.99^{\frac{1}{10}}$ |
| Target network update rate $\epsilon$ | $5e-3$ |
| Target divergence $\delta$ | 5 (Kitchen), 1 (Maze, Sparse Block Stacking), |
| **Variable Length Skill** | |
| Threshold of novelty | 0.3 (Kitchen), 50 (Maze), 40 (Sparse Block Stacking) |
| Maximum skill length $H$ | 30 |

Table 3: Training Hyperparameters

