# OpenReview forum: "Learning variable-length skills through Novelty-based Decision Point Identification"
_ICLR.cc/2024/Conference — Submitted to ICLR 2024_

### Official Review · Reviewer_Yx7H · 2023-10-29

**Soundness:** 2 fair
**Presentation:** 2 fair
**Contribution:** 2 fair
**Rating:** 3
**Confidence:** 4

**Summary:**

This paper studies learning variable-length skills from data to accelerate RL. It assumes that novel state-actions in data are critical dicision points and the low-level skills should terminate at such points. Thus, the paper proposes NBDI, training an ICM to estimate the state-action novelty and labeling the termination for each data sample. Then, the paper adopts the method of SPiRL, training a latent skill representation, an action decoder (with a termination decoder), and a skill prior, providing temporal abstractions for training a high-level policy with RL. Experiments in Maze show that the method outperforms SPiRL.

**Strengths:**

1. Extracting skills with variable lengths is beneficial for hierarchical RL. The paper points out a significant issue that existing methods which learn skills from data only use fixed-length skills.

2. Terminating skills at novel state-actions is an interesting attempt to address this problem. Experiments in Maze show that the proposed method outperform the method without variable-length skills (SPiRL); In robotic simulations, it slightly outperforms SPiRL.

**Weaknesses:**

1. The assumption that "novel state-actions are critical decision points" is **too strong**. I find that datasets used in this paper just meet this assumption: in Maze, the behavior policy only chooses diverse actions at crossroads; in Kitchen, the dataset consists of expert policies accomplishing a fixed set of skills, making the actions diverse at skill success points only. If we use other environments (e.g., robotic tasks without clearly defined skills) or non-expert data collection policies with stochasticity, this assumption will no longer hold, and it is doubtful whether NBDI can outperform SPiRL.

2. In implementations, the ICM is trained and evaluated on a fixed dataset, which **may not reflect the notion of "novelty"** discussed in the paper. In the original usage of ICM, it can estimate novelty in online RL, since the model cannot predict unseen transitions. However, this paper uses ICM to estimate novelty for the training data. Assuming that the model can overfit the dataset, the ICM losses on all training samples are small, thus ICM can fail to evaluate the state-action novelty in the dataset.

3. According to the context, the presentation of the Theorem in Section 4.2 seems redundant. The theorem tells that breaking skills into shorter pieces can improve the high-level controller. But the paper studies how to determine the skill termination points, instead of breaking pre-defined fixed-length skills into shorter pieces. Also, the conclusion in the theorem is straight-forward, while the detailed notations (Q and V) in it are no longer used in the paper. I think the the theorem should be moved to the Appendix.

4. The method is incremental to SPiRL. Its improvement on the two robotic simulation tasks is small, according to Figure 6.

**Questions:**

1. As I discussed in the Weaknesses (1), when the behavior policies in dataset are non-expert and stochastic, it is questionable whether NBDI can still provide benefits. I think more experimental results are required; otherwise, the authors should make this assumption and limitation clear in the paper.

2. In the experiments, how did you adjust the model capacity and training steps of ICM? If ICM overfits on training data, can it still represent state-action novelty, as I discussed in Weakness (2)?

3. When generating termination labels for the dataset, how to convert the continuous prediction loss of ICM into binary termination labels? Does this require selecting a threshold manually?

---

> ### Author Response · Authors · 2023-11-22
> **Response to Reviewer Yx7H**
>
> Dear Reviewer Yx7H,
>
> We thank the reviewer for the thorough and constructive comments. We hope we can address your concerns below.
>
> **1. Its improvement on the two robotic simulation tasks is small, according to Figure 6.**
>
> Thank you for pointing out this. We've incorporated Table 1 to highlight the degree of quantitative improvement, especially in complex environments, compared to SPiRL[2]. Please refer to the general response to see the details.
>
> **2. In robotic tasks without clearly defined skills it is doubtful whether NBDI can outperform SPiRL.**
>
> Thank you for your insightful feedback. In the sparse block stacking environment, the agent needs to stack blocks on top of each other. Since the agent’s sole objective is to stack as many blocks as possible, this environment can be regarded as a robotic task without clearly defined skills (whereas kitchen has clearly defined subtasks). Please refer to Appendix B for more details.
>
> Table 1 shows that NBDI outperforms SPiRL in the sparse block stacking environment (robotic tasks without clearly defined skills) with an improvement rate of 67.16%. Note it is a higher improvement rate compared to robotic task with clearly defined skills (kitchen: 22.33%). The significance of learning variable length skills with state-action novelty increases in environments lacking clearly defined subtasks (sparse block stacking).
>
> **3. With non-expert data collection policies with stochasticity, it is doubtful whether NBDI can outperform SPiRL.**
>
> Thank you for your constructive advice. We've added a new section (Appendix C) to investigate the extent to which our proposed method depends on the quality of the offline dataset. Please refer to the general response to see the details.
>
> **4. How did you adjust the model capacity and training steps of ICM[1]? If ICM overfits on training data, can it still represent state-action novelty, as I discussed in Weakness (2)?**
>
> Thank you for raising this important point. We have included a section (Appendix E) to show that the estimated state-action novelty is not much influenced by the number of parameters or dataset used to train the model. We also doubled the number of training steps, but observed that the scale didn't differ significantly.
>
> **5. When generating termination labels for the dataset, how to convert the continuous prediction loss of ICM into binary termination labels? Does this require selecting a threshold manually?**
>
> Thank you for pointing out this. Empirically, we discovered that state-actions falling within the top 1% of the prediction error percentile serves well as critical decision points, and used the corresponding threshold to learn the termination distribution.
>
> **6. The theorem tells that breaking skills into shorter pieces can improve the high-level controller. But the paper studies how to determine the skill termination points, instead of breaking pre-defined fixed-length skills into shorter pieces.**
>
> SPiRL basically extracts random fixed-length action sequences from the offline dataset by regarding the fixed-length action sequence in the dataset as pre-defined skills. On the other hand, our approach tries to break long action sequences using state-action novelty to learn better defined skills. Given that we are able to learn optimal skill (high-level) policy, the optimal way of defining skill would be using low-level actions directly as skills. In practice, however, shorter skills will increase the difficulty of learning, leading to poor skill policy. This is why we need a more clever and efficient break down of skills that resembles terminations with optimal improvement as shown in the Theorem [3] and the Figure 4, while obtaining similar temporal abstraction.
>
> [1] Deepak Pathak, Pulkit Agrawal, Alexei A Efros, and Trevor Darrell. Curiosity-driven exploration by self-supervised prediction. In International conference on machine learning, pp. 2778–2787. PMLR, 2017
>
> [2] Karl Pertsch, Youngwoon Lee, and Joseph Lim. Accelerating reinforcement learning with learned skill priors. In Conference on robot learning, pp. 188–204. PMLR, 2021a
>
> [3] Richard S Sutton. Between mdps and semi-mdps: Learning, planning, and representing knowledge at multiple temporal scales. 1998

---

### Official Review · Reviewer_Pqwe · 2023-10-30

**Soundness:** 2 fair
**Presentation:** 2 fair
**Contribution:** 2 fair
**Rating:** 5
**Confidence:** 3

**Summary:**

The paper presents an approach to for learning variable-length skills by detecting decision points based on the state-action novelty. The algorithm is built based on SPiRL in the offline reinforcement learning setting. The proposed approach achieves positive results on a navigation task and two simulated manipulation tasks for down-streaming evaluation of the learned skills.

**Strengths:**

1. The proposed approach to find critical decision point during skill discovery makes sense and is an important problem to study in skill/option discovery.

2. The paper is generally well-written and easy to understand. Source code is provided.

**Weaknesses:**

My main concern for this paper is with the empirical evaluations.
1. Lack of option discovery/learning baselines. As the authors also mentioned in the introduction section, option framework and its related algorithms typically learn a termination condition which functions as a tool to determine whether the agent needs to switch to a different skill. Several recent papers [1, 2, 3] propose to do skill/option discovery from offline data while learning a flexible termination condition just like this paper. But none of them are compared in the empirical results or discussed in the related work section. The proposed approach is only compared to SPiRL which uses a fixed skill length.

2. As it's a offline RL test setting, I think standard offline RL algorithms should be compared to instead of SAC. Moreover, as a general skill discovery method, I would suggest the authors test the proposed approach on more domains.

[1]. Learning options via compression. Neurips 2022.
[2]. Learning robot skills with temporal variational inference. ICML 2020.
[3]. Opal: Offline primitive discovery for accelerating offline reinforcement learning. ICLR 2021.

**Questions:**

1. Figure 4 is a little confusing. Different training objectives lead to different termination improvement occurrences, but is one of the three results better than the other two? It's not quite straight-forward to me.
2. In section 4.2, the authors claim "The termination improvement theorem basically implies that we should terminate an option when
there are much better alternatives available from the current state." So how do the authors decide which option is a better alternative in this paper's scope?

---

> ### Author Response · Authors · 2023-11-22
> **Response to Reviewer Pqwe**
>
> Dear Reviewer Pqwe,
>
> We appreciate your comprehensive comments. Please find the response to your questions below.
>
> **1. Lack of option discovery/learning baselines**
>
> We appreciate your constructive advice for the experiment. We added a section (Appendix F) to compare NBDI with different variable length algorithm. Please refer to Appendix F for more details.
>
> We also attempted to incorporate [2] into the comparison; however, we were unable to replicate the results reported in the paper since the dataset used to train [2] was not made available. So we mainly compared our model with [3] in Appendix F.
>
> **2. Standard offline RL algorithms should be compared to instead of SAC**
>
> As depicted in Figure 5, both NBDI and SPiRL[1] need environment interactions to train the skill policy. As our paper primarily focuses on the online RL setting, we believe SAC serves as a suitable baseline for comparison across various environments.
>
> **3. Test the proposed approach on more domains.**
>
> Thank you for this suggestion. We've incorporated a section (Appendix D) where we explore the possibility of identifying meaningful decision points in complex physics simulation environments. Please refer to Appendix D for more details.
>
>
> [1] Karl Pertsch, Youngwoon Lee, and Joseph Lim. Accelerating reinforcement learning with learned skill priors. In Conference on robot learning, pp. 188–204. PMLR, 2021a
>
> [2] Shankar, Tanmay, and Abhinav Gupta. "Learning robot skills with temporal variational inference." *International Conference on Machine Learning*. PMLR, 2020.
>
> [3] Jiang, Yiding, et al. "Learning Options via Compression." *Advances in Neural Information Processing Systems* 35 (2022): 21184-21199.

---

### Official Review · Reviewer_zT8h · 2023-10-31

**Soundness:** 4 excellent
**Presentation:** 3 good
**Contribution:** 3 good
**Rating:** 8
**Confidence:** 3

**Summary:**

The paper proposes a unsupervised skill learning algorithm that can extract variable-length skills from a task-agnostic offline dataset. Variable-length skills are motivated by the observation that fixed-length skills often move past critical decision points, like crossroads, resulting in suboptimal performance on downstream tasks. The authors propose to detect such decision points using state-action novelty. The resulting algorithm, Novelty-based Decision Point Identification (NBDI), learns skills together with their termination probability and a prior over the skill space using a latent variable model similar to SpiRL [1]. The authors furthermore provide a complementary perspective on variable termination of skills by relating it to the potential performance gains achieved by terminating and switching to higher-value options, arguing that state-action novelty serves as a proxy for situations where this is likely to be beneficial. Experiments in two maze environments and two robotic manipulation tasks demonstrate that the variable-length skills learned by NBDI lead to better performance in downstream tasks.

[1] Karl Pertsch, Youngwoon Lee, and Joseph Lim. Accelerating reinforcement learning with learned skill priors. In Conference on robot learning, pp. 188–204. PMLR, 2021.

**Strengths:**

The need for variable-length skills is well motivated by practical examples and a more theoretical argument based on the potential benefits of early option termination.

The paper is furthermore well written and structured and easy to follow. The figures generally do a good job in conveying the intuition behind the algorithm as well as in illustrating the three phases of the algorithm.

The variations of NBDI with different kinds of novelty signals for the training of the termination probability are a good addition to the experiments as they demonstrate that considering state-action novelty is crucial.

The authors furthermore already made the code public.

**Weaknesses:**

The algorithm is motivated with the concept of novelty, explicitly using a count-based notion of novelty in equation (1). In the experiments section the paper then briefly mentions that Intrinsic Curiosity Module (ICM) is used to obtain state-action novelty values. However, ICM obtains its intrinsic motivation signal from the prediction error of the next state in a learned feature space which is conceptually somewhat different. I think it would be a good idea to discuss to which extent the exact nature of the curiosity signal influences the learned skills.

As evident in the motivating examples, the learned skill termination probabilities depend on the data, in particular on where different actions have been chosen frequently. This poses the question of how dependent NBDI is on a suitable structure being present in the offline dataset. For example, could NBDI work with exploratory data, and if yes, under which circumstances? How does the complexity of the environment influence the requirements on the data? A discussion of these questions would be a good addition to the paper.

Overall the algorithm is fairly close to SpiRL but this is made transparent so it is not really a problem.

The additional ablations in Appendix A are quite interesting, in particular, the study of the novelty threshold. It would therefore be good to mention them more explicitly in the main text.

**Questions:**

* In the paragraph “Option Framework”, in the definition $\beta: \mathcal{S}^+ \rightarrow [0, 1]$ the symbol $\mathcal{S}^+$ is not defined anywhere. What does it stand for?
* In equation (5) $\tilde{r}$ is added to the objective in each time step even though $\tilde{r}$ itself is already the cumulative discounted reward for the execution of one skill. Could you maybe explain how to reconcile these definitions?

---

> ### Author Response · Authors · 2023-11-22
> **Response to Reviewer zT8h**
>
> Dear Reviewer zT8h,
>
> We thank the reviewer for the detailed feedback. We address your remarks below.
>
> **1. To which extent the exact nature of the curiosity signal influences the learned skills.**
>
> We acknowledge the concern raised regarding the missing details, and we would like to address this by providing the necessary information in Appendix B. Please refer to the general response to see the details.
>
> **2. How dependent NBDI is on a suitable structure being present in the offline dataset. For example, could NBDI work with exploratory data, and if yes, under which circumstances?**
>
> We appreciate your constructive advice for the experiment. We've introduced a new section (Appendix C) to explore to what extent our proposed method is depend on the quality of the offline dataset. Please refer to the general response to see the details.
>
> **3. How does the complexity of the environment influence the requirements on the data?**
>
> We have also added a section (Appendix D) to investigate whether meaningful decision points can be found in complex physics simulation environments. Please refer to Appendix D for more details.
>
> **4. In the paragraph “Option Framework”, in the definition $\beta:S^+\to[0,1] $ the symbol  is not defined anywhere. What does it stand for?**
>
> We apologize for the confusion caused. We initially intended to define $S^+$ as the termination set in accordance with [2], but we deemed it unnecessary and substituted it with the state space $S$.
>
> **5. In equation (5) $\tilde{r}$ is added to the objective in each time step even though $\tilde{r}$ itself is already the cumulative discounted reward for the execution of one skill. Could you maybe explain how to reconcile these definitions?**
>
> Thank you for pointing this out. To reconcile the definition, we have now defined a set of time steps where we execute skills (Equation 5).
>
> [1] Karl Pertsch, Youngwoon Lee, and Joseph Lim. Accelerating reinforcement learning with learned skill priors. In Conference on robot learning, pp. 188–204. PMLR, 2021a
>
> [2] Richard S Sutton. Between mdps and semi-mdps: Learning, planning, and representing knowledge at multiple temporal scales. 1998

---

> > ### Comment · Reviewer_zT8h · 2023-11-22
> >
> > I thank the reviewers for the answers to my questions and the additional experiments included in the revision of the paper. I appreciate the work that went into the additional sections.
> >
> > 1. Thank you for the clarifications with respect to the use of ICM. Two quick questions about this: As you train on the same transitions you evaluate the model on, how do you make sure that you are not overfitting to the dataset? There is furthermore one sentence I did not quite understand in Appendix B: "low prediction error occurs in states where we do not have multitude of plausible actions, or rare configuration ((D), (E), (F)" Why does low prediction error correspond to rare configurations?
> >
> > 2. The experiments with a lower quality, noisy policy are interesting. It looks like NBDI is able to cope with such data to some extent so this is a good addition.
> >
> > 3. The example decision points with the Ant being able to flip or stabilize on its legs makes a lot of sense, thank you. The other decision points are a bit harder to interpret from images. A video might help with that. It makes sense that completely random data does not result in interpretable or useful decision point and will therefore have a negative impact on the performance of NBDI. As long as this is made transparent in the paper, this is fine, I think.
> >
> > 4. Thank you!
> >
> > 5. Ok, great. I think the equation is clear now.

---

> > > ### Author Response · Authors · 2023-11-23
> > > **Re: Official Comment by Reviewer zT8h**
> > >
> > > We thank the reviewer for reading our rebuttal. Please find the response to your further questions below.
> > >
> > > **1. As you train on the same transitions you evaluate the model on, how do you make sure that you are not overfitting to the dataset?**
> > >
> > > We have incorporated a section (Appendix E) to demonstrate that the estimated state-action novelty remains relatively unaffected by the model's parameter count or the dataset used for training. We also doubled the number of training steps, but observed that the scale didn't differ significantly. So ICM shows some robustness to changes in the number of parameters, training steps, and the size of datasets used for training when employed as a state-action novelty estimator.
> > >
> > > **2. Why does low prediction error correspond to rare configurations?**
> > >
> > > We apologize for any confusion. As depicted in Figure 10 and Figure 13, states with rare configurations correspond to a high prediction error (the red box denotes the goal in the maze environment, which is sparsely represented in the dataset).

---

> > > > ### Comment · Reviewer_zT8h · 2023-11-23
> > > >
> > > > Great, thank you! This answers my questions.

---

### Official Review · Reviewer_6rC9 · 2023-11-14

**Soundness:** 3 good
**Presentation:** 2 fair
**Contribution:** 2 fair
**Rating:** 3
**Confidence:** 4

**Summary:**

The paper proposes an approach to learn variable length skills in RL, closely building on SpiRL [1]. This is done using state-action novelty to predict a termination condition for the skills.








[1] - Pertsch, Karl, Youngwoon Lee, and Joseph Lim. "Accelerating reinforcement learning with learned skill priors." Conference on robot learning. PMLR, 2021

**Strengths:**

RL with skills can enable much more efficient learning, and is an important problem. Learning variable length skills should allow for more effective learning. However, I have some concerns regarding this paper - please see weaknesses.

**Weaknesses:**

1. Clarity of proposed approach

The method of variable length skills relies heavily on using state-action novelty to prediction termination conditions. However the exact details for how the state-action novelty module is trained is not explained in the paper (box (i) in the method figure). The paper includes some description of characterizing novelty as inverse visitation count, so do the authors explicitly maintain counts of each seen state? (This will be difficult to scale to environments with continuous states and actions). Or is there some other metric of pseudo counts, involving density estimation? What is the relative effect of using different means to estimate the novelty metric, to predict termination? The reason for this particular analysis is that the novelty estimation is the main contribution of this paper.

2. Significance of contribution

The proposed method heavily builds on prior work [1], for the components that learn skills and perform RL in the skill space. From the experimental results, the quantitive gains over [1] seem very marginal in the more complex environments (sparse block stacking, kitchen). Can the authors include some analysis of how their discovered skills differ qualitatively in these more complex envs, as compared to [1]?. What are the specific cases in these environments that variable length skills are actually helpful/needed?


[1] - Pertsch, Karl, Youngwoon Lee, and Joseph Lim. "Accelerating reinforcement learning with learned skill priors." Conference on robot learning. PMLR, 2021

**Questions:**

Please address questions in the weaknesses section

---

> ### Author Response · Authors · 2023-11-22
> **Response to Reviewer 6rC9**
>
> Dear Reviewer 6rC9,
>
> Thank you for the valuable feedback. We hope we can address your concerns below.
>
> **1. Clarity of proposed approach (state-action novelty estimation)**
>
> We apologize for the missing details. We've included a section (Appendix B) that details our application of the Intrinsic Curiosity Module (ICM)[1] in calculating state-action novelty. Please refer to the general response to see the details.
>
> **2. The quantitive gains over SPiRL[2] seems very marginal in the more complex environments (sparse block stacking, kitchen).**
>
> Thank you for pointing out this. We've included Table 1 to highlight the extent of quantitative improvement over SPiRL (including complex environments). Please refer to the general response to see the details.
>
> **3. Can the authors include some analysis of how their discovered skills differ qualitatively in these more complex environments, as compared to [2]? What are the specific cases in these environments that variable length skills are actually helpful/needed?**
>
> Thank you for the valuable suggestion. We've included Figure 11 (Appendix B) to illustrate states in the sparse block stacking environment that NBDI would learn to terminate (A, B, C) and states it would not learn to terminate (D, E, F). NBDI would learn to terminate in states where the agent has multitude of plausible actions (A, B, C) (e.g. choose to lift the block below or to move towards other blocks), while SPiRL would terminate in random states (A, B, C, D, E, F). As indicated by the improvement rate in Table 1, the significance of learning variable length skills with state-action novelty becomes increasingly crucial in environments lacking clearly defined subtasks (sparse block stacking), or in more complex settings (maze 40x40).
>
> [1] Deepak Pathak, Pulkit Agrawal, Alexei A Efros, and Trevor Darrell. Curiosity-driven exploration by self-supervised prediction. In International conference on machine learning, pp. 2778–2787. PMLR, 2017
>
> [2] Karl Pertsch, Youngwoon Lee, and Joseph Lim. Accelerating reinforcement learning with learned skill priors. In Conference on robot learning, pp. 188–204. PMLR, 2021a

---

### Author Response · Authors · 2023-11-22
**General Response**

We leave the general response to the questions that are commonly mentioned.

**1. State-action novelty estimation with Intrinsic Curiosity Module(ICM)[1]**

We have added a section(Appendix B) to describe how we used Intrinsic Curiosity Module (ICM) to calculate state-action novelty.

While ICM is commonly known for providing intrinsic motivation signals to stimulate exploration in online RL, it is also treated as a effective state-aciton novelty estimator simliar to RND[2] since ICM would have high prediction error for rare state-action pairs among the state-action pairs it have seen.

Figures 10 and 11 in the appendix indicate that states with a multitude of plausible actions or rare configurations often exhibit high prediction errors. It's important to note that these characteristics correspond to the conditional action novelty and state novelty, as illustrated in Figure 4, leading to a high level of state-action novelty as described in Equation 1. On the other hand, states that lack these properties display a low prediction error, as depicted in Figure 10 and Figure 11. Moreover, our experiments in Appendix E demonstrate that the estimated state-action novelty is not much influenced by the number of parameters or the dataset used for training the model. Therefore, our results indicate that the Intrinsic Curiosity Module (ICM) can effectively function as a state-action novelty estimator when pre-trained with offline trajectory datasets. Furthermore, we found that state-actions within the top 1% prediction error percentile serve well as a critical decision points, and used the corresponding threshold to learn the termination distribution.

**2. Improvement over SpiRL[3] in complex environments (sparse block stacking, kitchen)**

We have included a table (Table1) that illustrates the extent of quantitative improvement over SpiRL. In a robotic simulation environment that has clearly defined subtasks (kitchen), NBDI outperforms SpiRL with an improvement rate of 22.33%. However, in a more challenging robotic simulation environment that has no clearly defined subtasks (Sparse block stacking), NBDI outperforms SpiRL with an improvement rate of 67.16%. As the environment becomes more complex and challenging, NBDI outperforms SpiRL with a higher improvement rate, just like the maze environment (Improvement over SpiRL in Maze 30x30: 84.62%, Maze 40x40: 177.78%). We believe that the improvement rates are significant although it was not clearly visible from the previous figures.

**3. Can NBDI work with non-expert level offline datasets with stochasticity?**

We have added a section (Appendix C) to investigate how dependent NBDI is on the quality of the offline dataset. We trained a behavior cloning (BC) policy using expert-level trajectories in the maze environment to generate demonstrations of mediocre quality. Additionally, we introduced weighted Gaussian random noise to the actions of the BC policy to give some stochasticity to the generated dataset. Table 2 in Appendix C illustrates that, even in a less challenging goal-setting scenario compared to Figure 7, SPiRL falls short of reaching the goal, whereas NBDI attains success rates of 28% and 22%. Figure 12 illustrates that SPiRL is limited to navigating around the initial state using fixed-length skills extracted from the suboptimal dataset, while NBDI successfully reaches the goal by employing variable-length skills. Figure 13 (top, middle) indicates that, even with a suboptimal dataset, NBDI can still learn termination points characterized by high conditional action novelty or state novelty. However, when utilizing a dataset generated through random walk (Figure 13 (bottom)), it becomes challenging to learn meaningful decision points. As the policy generating the trajectory becomes more stochastic, it collects data primarily around the initial state, resulting in an overall decrease in the magnitude of prediction errors. Thus, we can see that the level of stochasticity influences critical decision point detection, and NBDI might not work when the offline dataset is collected by a random walk policy.

[1] Deepak Pathak, Pulkit Agrawal, Alexei A Efros, and Trevor Darrell. Curiosity-driven exploration by self-supervised prediction. In International conference on machine learning, pp. 2778–2787. PMLR, 2017

[2] Burda, Yuri, et al. "Exploration by random network distillation." *arXiv preprint arXiv:1810.12894* (2018).

[3] Karl Pertsch, Youngwoon Lee, and Joseph Lim. Accelerating reinforcement learning with learned skill priors. In Conference on robot learning, pp. 188–204. PMLR, 2021a

---

### Meta-Review · Area_Chair_NeT1 · 2023-12-09

**Metareview:**

This paper studies discovering variable-length skills for hierarchical RL. Its algorithm NBDI leverages the changes of state-action pairs in data by assuming they are critical decision points and the low-level skills should terminate at those points. While approach seems novel, reviewers have the concerns on the limitation of the assumption of novel state-action pairs in data, implementation does not utilize the key assumptions of novel state-action pairs, the scale of experiments being too small, and redundant theoretical presentations.

**Justification For Why Not Higher Score:**

Concerns raised by reviewers needed to be addressed before paper can be considered for acceptance.

**Justification For Why Not Lower Score:**

N/A

---

### Decision · Program_Chairs · 2024-01-16

Reject